# Mid-infrared emissivity of partially dehydrated asteroid (162173) Ryugu shows strong signs of aqueous alteration

M. Hamm [1,2 ✉], M. Grott [2], H. Senshu[3], J. Knollenberg[2], J. de Wiljes [1], V. E. Hamilton [4], F. Scholten[2], K. D. Matz[2], H. Bates[5,6], A. Maturilli[2], Y. Shimaki [7], N. Sakatani [8], W. Neumann [2,9], T. Okada [7], F. Preusker[2], S. Elgner [2], J. Helbert[2], E. Kührt [10,11], T.-M. Ho[12], S. Tanaka[7], R. Jaumann[13] & S. Sugita [14]

The near-Earth asteroid (162173) Ryugu, the target of Hayabusa2 space mission, was observed via both orbiter and the lander instruments. The infrared radiometer on the MASCOT lander (MARA) is the only instrument providing spectrally resolved mid-infrared (MIR) data, which is crucial for establishing a link between the asteroid material and meteorites found on Earth. Earlier studies revealed that the single boulder investigated by the lander belongs to the most common type found on Ryugu. Here we show the spectral variation of Ryugu's emissivity using the complete set of in-situ MIR data and compare it to those of various carbonaceous chondritic meteorites, revealing similarities to the most aqueously altered ones, as well as to asteroid (101955) Bennu. The results show that Ryugu experienced strong aqueous alteration prior to any dehydration.

[1] Institute of Mathematics, University of Potsdam, Potsdam, Germany. [2] Institute of Planetary Research, German Aerospace Center (DLR), Berlin, Germany. [3] Planetary Research and Exploration Center, Chiba Institute of Technology, Narashino, Japan. [4] Southwest Research Institute, Boulder, CO, USA. [5] Department of Earth Sciences, Natural History Museum, Cromwell Road, London SW7 5BD, UK. [6] Atmospheric, Oceanic and Planetary Physics, Oxford University, Parks Road, Oxford OX1 3PU, UK. [7] Institute of Space and Astronautical Science, Japan Aerospace Exploration Agency, Sagamihara, Japan. [8] Department of Physics, Rikkyo University, Toshima, Japan. [9] Klaus-Tschira-Labor für Kosmochemie, Institut für Geowissenschaften, Universität Heidelberg, Im Neuenheimer Feld 234-236, 69120 Heidelberg, Germany. [10] Institute of Optical Sensor Systems, German Aerospace Center, Berlin, Germany. [11] Qian Xuesen Laboratory of Space Technology, China Academy of Space Technology, Beijing, China. [12] German Aerospace Center (DLR), Institute of Space Systems, Bremen, Germany. [13] Freie Universität Berlin, Berlin, Germany. [14] University of Tokyo, Tokyo, Japan. ✉email: maximilian.hamm@dlr.de

The JAXA Hayabusa2 mission to (162173) Ryugu investigated the asteroid's surface from 2018–2020, and successfully returned samples to Earth in December 2020[1–3]. Based on the analysis of colour ratios in the Optical Navigation Camera (ONC[4]) images and near-infrared (near-IR) spectra taken by the Near-Infrared Spectrometer (NIRS3[5]), Ryugu seems to be predominantly composed of material similar to carbonaceous chondrites (CC) that were heated after experiencing aqueous alteration[6–8]. In particular, the NIRS3 data show the ubiquitous presence of Mg-rich phyllosilicates as indicated by a sharply defined 2.72-μm absorption band, yet the absence of any band at slightly longer wavelengths near 3.00 μm indicate no other water-bearing minerals such as smectite phyllosilicates were found[9]. The thermal infrared mapper (TIR[10]) measured the surface temperature by observing the 8–12 μm spectral region determining the thermal inertia of the surface[11]. In addition to the global remote-sensing investigation of Ryugu, Hayabusa2 deployed the MASCOT (Mobile Asteroid Surface Scout) lander to the asteroid's surface[12]. MASCOT investigated a single boulder on Ryugu in situ. Camera images revealed that this boulder belongs to the most common type of boulders on Ryugu[13].

Among the instruments onboard MASCOT, the thermal infrared radiometer (MARA) observed the infrared emission of the boulder's surface in six channels with different filter windows. Two long-pass channels were designed to measure the brightness temperature of the boulder and derive its thermal inertia, i.e., the material parameter governing the speed and extent of the temperature change induced by insolation. One of these broadband filters (SiLP) covers the range from 3.5–100 μm, and the second one (W10) is identical to the main spacecraft's TIR filter, covering 8–12 μm. The four narrow-band channels were designed to estimate the surface emissivity within 5.5–7 μm (B06), 8–9.5 μm (B08), 9.5–11.5 μm (B09), and 13.5–15.5 μm (B13)[14]. MARA provides the only multi-spectral data in the mid-infrared (mid-IR), besides telescopic, disk-integrated observations[15,16], complementing the main spacecraft remote observations while at the same time observing a pristine boulder surface.

The mid-infrared wavelength region of the electromagnetic spectrum shows characteristic minima and maxima that are a function of the composition and structure of organic and inorganic materials[17]. These features allow to identify possible rock compositions on the surface of Ryugu. In silicate minerals and silicate-dominated rocks, an emissivity peak known as the Christiansen feature (CF), is located between 7.5 and 9 μm and its position is diagnostic of the polymerisation of the silicate structure[18] and indicative of the bulk composition of minerals

and rocks. In silicates, bending and stretching vibrational modes produce diagnostic absorption bands in the ~8–12 and 15–30 μm region[18–23]. In the region between 4 and 7 μm, some minerals, such as pyroxene or olivine can show weaker absorptions corresponding to overtones of molecular or lattice vibrations and carbonate minerals have a fundamental absorption in this region[22]. Furthermore, molecular water shows characteristic absorption bands in that region[17,24].

The mid-IR spectrum also depends on physical conditions. As the particle size becomes comparable to the wavelength of the measurement, the contrast of the vibrational modes is reduced and new features, referred to as transparency features, appear due to volume scattering. In silicates, transparency features occur at wavelengths shorter than the CF and in the interband region between ~11 and 13 μm as well as at longer wavelength[25–29]. In addition, in fine, transparent particulate materials, steep thermal gradients can develop in the upper few 10–100 s of μm and increase spectral contrast due to the measurement of multiple temperatures from different depths of the sample simultaneously. This effect is enhanced under very low pressure and vacuum conditions[18,30]. However, for low-albedo materials, including carbonaceous chondrites containing opaques and insoluble organic material, this effect is substantially less pronounced[22,31–33]. On Ryugu, with its dark and dust-deficient surface, volume scattering and thermal gradients should contribute little to the observed emissivity.

Previous analysis of the MARA data used W10 filter data (8–12 μm band) to reveal a very low bulk thermal inertia of 247–375 J m$^{-2}$ K$^{-1}$ s$^{-1/2}$, which is likely caused by a very high porosity of 28–55%[34,35]. Such high porosities are only found in the most porous of CM and CI chondrites[34,36]. In these works, the unknown orientation of the observed surface and the thermal radiation received from the surrounding terrain had to be included as free parameters in the analysis which increased the uncertainty of the emissivity estimate within the narrow-band filters to a degree that hindered a meaningful interpretation. With the derivation of a detailed 3D shape model of the boulder based on MASCOT camera images[37] (Fig. 1), the orientation of the observed surface, as well as its radiative exchange with the local terrain, can be considered fixed and the analysis of the full MARA dataset is feasible.

Here, we present the analysis of the full MARA dataset, including all six instrument channels. The boulder shape model is combined with a digital elevation model (DEM) of the region around MASCOT's landing site and coupled to a thermophysical model (Fig. 1c and section "Thermal model of the MASCOT landing site" under "Methods"). Applying an Ensemble Kalman

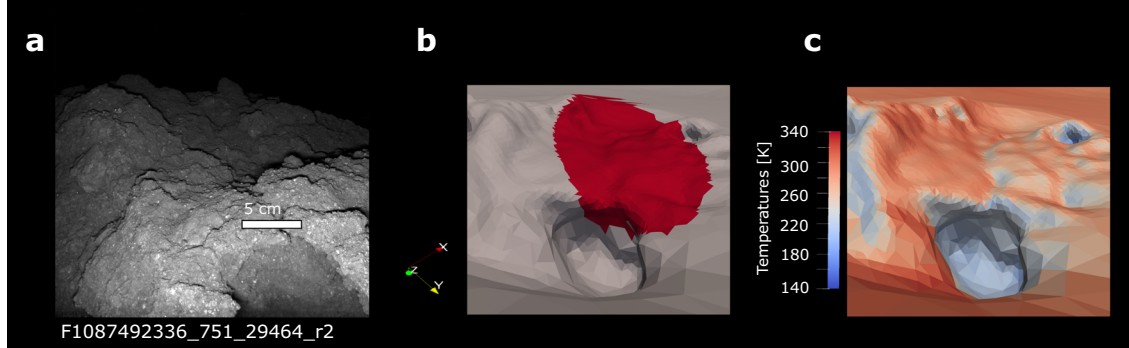

**Fig. 1 Images and DEM of the observed boulder. a** MASCAM image of the boulder taken during night illuminated by the red LED array of MASCAM[13]. The text below the image denotes the image number. The boulder is of the common, dark, rugged type of boulders on Ryugu. **b** The DEM of the boulder[37] with the combined field of view of the MARA instrument's six channels shown in red. Axis show the orientation of the DEM within the asteroid's body-fixed frame for panels **b** and **c**. **c** Same scene as in **b** overlaid by a snapshot of the thermal model calculation showing the heterogenous temperatures within the MARA field of view due to the boulder roughness.

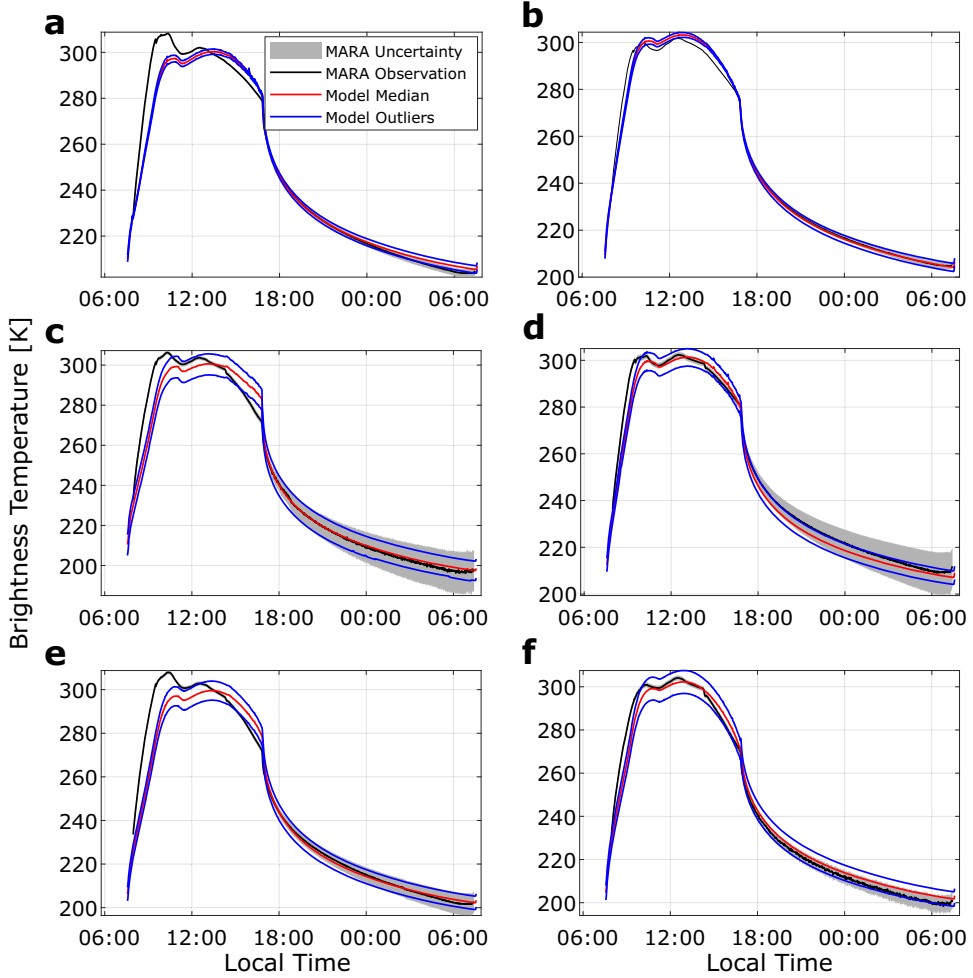

**Fig. 2 Observed brightness temperature of all six instrument channels as a function of local time in comparison to modelled observation. a** Brightness temperature (black) observed in the Silicon-Long-pass (SiLP) is shown as a function of local time (given in hours, defined by 1/24 of Ryugu's rotation period) along with the corresponding uncertainty (2σ with σ the standard deviation) given by grey shades. Red line indicates the median of modelled observation based on the posterior parameter estimation. Blue lines are models based on the range of outliers of the posterior within 1.5 times the interquartile range from the 25th and 75th percentile, respectively. **b–f** As for a but for the other instrument channels, **b** 8–12 µm filter (W10), **c** 5.5–7 µm (B06), **d** 8–9.5 µm (B08), **e** 9.5–11.5 µm (B09) and **f** 13.5–15.5 µm (B13). Source data are provided as a Source Data file.

filter (EnKF, see the section "Ensemble Kalman filter" under "Methods") for parameter estimation, we retrieve the thermal inertia and emissivity of the boulder's material, which we compare to the spectra of various carbonaceous chondrites. We find that the emissivity of Ryugu is similar to the most aqueously altered carbonaceous CI and CM chondrites, indicating that Ryugu experienced substantial aqueous alteration prior to any later dehydration of its surface. This analysis of the MARA data can be directly compared to the analysis of samples returned to Earth.

## Results

**Brightness temperature and thermal inertia**. The thermal inertia depends on the structure of the respective surface. Small particle size and high porosity of the material both reduce the thermal conductivity and with it the thermal inertia. Upon illumination, the surface heats up faster and to higher temperatures compared to more compact or coarse material. The observed brightness temperatures and the corresponding uncertainties of all six MARA channels are shown in Fig. 2, along with the best-fitting thermal models. The brightness temperature is a concept to facilitate the interpretation of observed infrared flux. It is the

temperature the surface would have if it were an ideal black-body radiator emitting the observed infrared flux. Although we show the plots in terms of brightness temperature for the sake of clarity, we fit the observed infrared flux directly in our parameter estimation method. The applied EnKF is a specific Monte-Carlo method in which random parameter combinations are iteratively updated until their histograms form a distribution, the posterior, which describes the uncertainty of the parameters given the observation and the model (see the section "Ensemble Kalman filter" under "Methods").

The results here and in the following are given by the median of the posterior parameter distribution. The uncertainties are defined by the first and third quartile (defined by the parameter value at which 25% and 75% of the retrieved posterior parameters are smaller, respectively). The total range of the parameters, including outliers, is shown in Fig. 3. It is approximately three times the range of the given uncertainty. The red curve in Fig. 2 shows the median of the modelled brightness temperature, based on the estimated parameters. The blue lines indicate the corresponding outliers. It is apparent that the thermal model cannot explain the entire dataset equally well. The best fit is achieved in the broadband W10 filter, which also has the lowest temperature uncertainty. The differences between observation

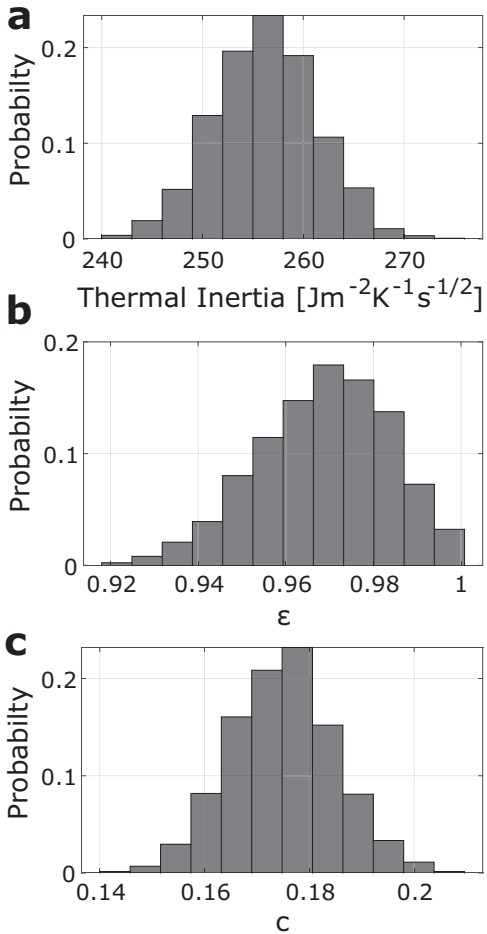

**Fig. 3 Estimated parameter distributions.** Histograms over the posterior parameter distributions for (**a**) thermal inertia, (**b**) broadband emissivity $\varepsilon$ (**c**) and crater density of the roughness model $c$, with $y$ axis representing the probability of a parameter value within the interval given by the histogram bars. Parameter combinations with emissivity >1 are discarded. The histograms are formed from the results of the data assimilation and represent the uncertainty of the parameter estimation according to the observation uncertainty. Source data are provided as a Source Data file.

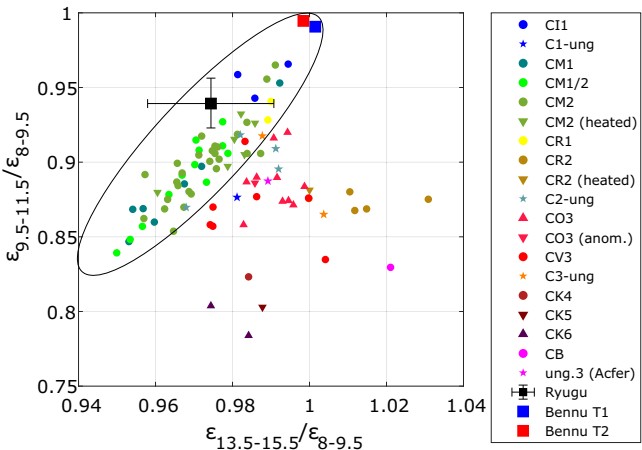

**Fig. 4 Comparison of emissivity ratios derived from MARA data to emissivity ratios of chondrite thin sections and Bennu.** Emissivity spectra of the meteorite samples are derived from inverted reflectance spectra of thin sections of carbonaceous chondrites. Black square with error bars shows the ratio of the estimated emissivity of Ryugu in the 8–9.5 μm, 9.5–11.5 μm, and the 13.5–15.5 μm bands with error bars indicating the 25th and 75th percentile of the estimated band ratios. Inverted reflectance spectra of thin sections of various carbonaceous chondrites[44] and Bennu's emissivity spectra, where T1 and T2 different spectral types of the surface[44], are averaged within the MARA channels and weighted by the MARA instrument functions. The aqueously altered CM, CI, CR1 chondrites, Bennu and Ryugu form a common trend, from which the CV, CO, CR2, CB, and CK deviate systematically. Chondrites that were subjected to heating (downwards pointing triangles) do no vary significantly from the unheated chondrites of the same type. The ellipse is shown to guide the eye. Source data are provided as a Source Data file.

and model are due to limitations of the boulder shape model, where part of the field of view lies towards the edges of the camera images and the accuracy of the derived shape model is low.

Furthermore, the temperatures between sunrise and noon are influenced by a reflection of sunlight from the MASCOT exterior into the field of view. Modelling this reflection is challenging and produces systematic uncertainties (see the section "MASCOT sunlight reflection" under Methods). Therefore, we only include observations into the analysis afternoon when the reflected sunlight passed out of the field of view of the instrument. To estimate the influence of systematic model uncertainties, we repeat the analysis for a model neglecting the reflected sunlight and that does not require sub-facet roughness to fit the data. Despite the two very different models, the results are similar and do not change the conclusion presented in this work. The results for the simplified model are summarised in Supplementary Note 1, Supplementary Figs. 1 and 2 along with a comparison to earlier studies[34], Supplementary Fig. 3, which are very similar to the reduced model. The more complex model results in better fits to the MARA data and is the focus of this work.

The thermal inertia of the boulder is estimated to be $256^{+4}_{-3}$ J m$^{-2}$ K$^{-1}$ s$^{-1/2}$. The corresponding posterior distribution is shown in Fig. 3. Such a thermal inertia is very low for rocks and millimetre-sized particles were expected to cover the surface of Ryugu based on similar thermal inertia estimates prior to the arrival of Hayabusa2[38]. The presence of a dust layer that might mask a higher thermal inertia of the underlying boulder with low thermal inertia can be excluded as such a layered surface shows a systematically different cooling rate as the one observed[34,39]. The observed cooling rate indicates that the rock has a high porosity[34,40]. The estimated thermal inertia corresponds to a porosity of $46.7^{+0.3}_{-0.4}\%$ (see the section "Porosity estimate based on thermal inertia" under "Methods"), which is more porous than carbonaceous chondrites found on Earth[41]. Although MASCOT observed only a single boulder, these results are representative for large parts of Ryugu as the boulder belongs to the most common, dark, rugged-shaped type of boulders on Ryugu's surface[7,13]. This is substantiated by comparing our thermal inertia estimate to the global distribution of thermal inertia estimates for individual boulders, which places our estimate among the most common of boulders[11,40,42].

**Mid-IR emissivity estimates.** The emissivity of the surface material determines its efficiency of emitting energy. The closer it is to 1, the better the surface is described by a blackbody. Estimates resulting in emissivity >1 are discarded from the analysis. The estimated emissivity in the broadband channels is high with $\varepsilon = 0.97^{+0.01}_{-0.01}$ and the corresponding posterior distribution is shown in Fig. 3. This emissivity is also assumed to be the total emissivity of the surface in the thermal model, determining the radiative cooling of the surface (see the section "MARA observation model" under "Methods"). Variations in the emissivity

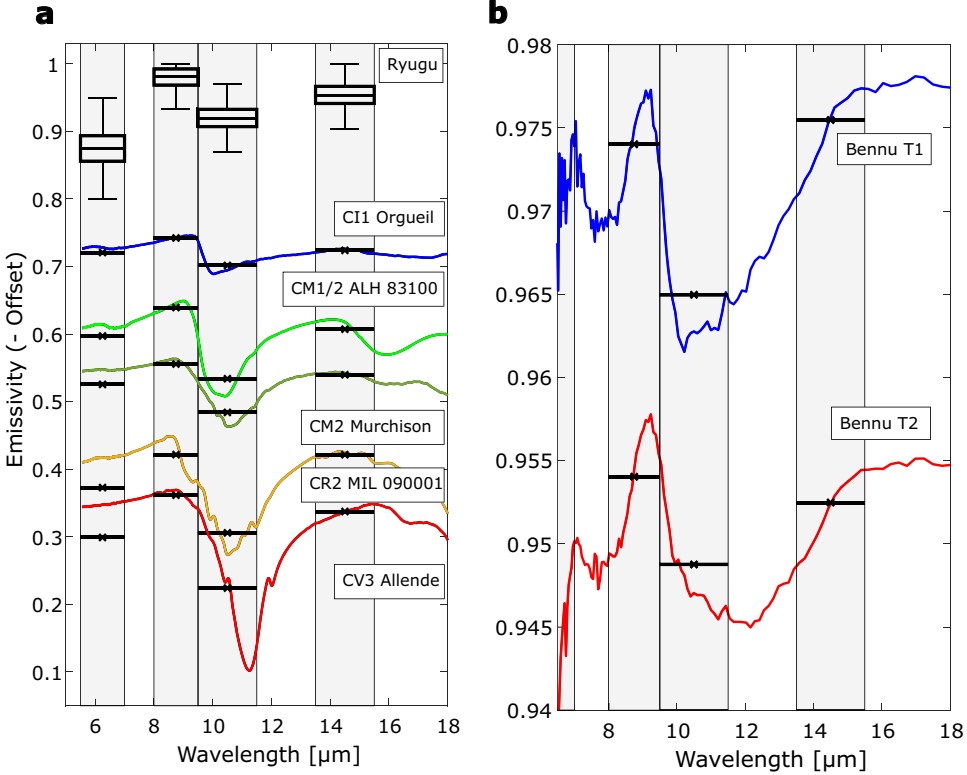

**Fig. 5 Comparison of derived emissivity of Ryugu compared to inverted reflectance spectra of thin sections of carbonaceous chondrites and OTES emissivity spectra of Bennu. a** Boxplot of the posterior distribution of the emissivity within each of the four narrow-band channels compared to spectra of examples of thin-section carbonaceous chondrites[44], where horizontal lines denote the spectra averaged within the MARA channel and weighted by the instrument function. Reflectance spectra are converted to emissivity by subtracting the reflectance from 1, and shifted by a constant offset for clarity. The box boundaries are given by the 25th and 75th percentile, the centre is defined by the median, the error bars indicate the range of outliers within 1.5 times the interquartile range from the 25th and 75th percentile, respectively. All thin sections show an emissivity drop in the 9.5–11.5 μm band and the general trend of increasing emissivity from the 5.5–7 μm band to the 8–9.5 μm band and from the 9.5–11.5 μm to the 13.5–15.5 μm band, yet the relative depth of the bands vary systematically. The colours of the spectra were chosen to match the corresponding symbols in Fig. 4. **b** Emissivity spectra of Bennu[44] along with corresponding averages in the MARA channels. Emissivity in the 5.5–7 μm is not averaged due to measurement artefacts within that wavelength range. The T1 spectrum is shown as measured, the T2 spectrum is shifted by 0.02. Source data are provided as a Source Data file.

spectrum will cause variations in the observed infrared flux if the instrument channel observes the surface at these specific wavelengths. The emissivity retrieved in the narrow-band filters shows similarities to aqueously altered, carbonaceous meteorites. The emissivity drops within the 5.5–7 μm band to $\varepsilon_{B06} = 0.87^{+0.02}_{-0.01}$, reaches its maximum in the 8–9.5 μm band with $\varepsilon_{B08} = 0.98^{+0.01}_{-0.01}$, drops in the adjacent 9.5–11.5 μm band to $\varepsilon_{B09} = 0.92^{+0.01}_{-0.01}$, and rises again to $\varepsilon_{B13} = 0.95^{+0.02}_{-0.01}$ in the 13.5–15.5 μm region.

We compare these results to mid-IR reflectance spectra of thin sections (Figs. 4 and 5)[43,44] and powdered carbonaceous chondrites[45,46] as well as emissivity spectra of powdered carbonaceous chondrites collected under simulated asteroid environment (SAE) conditions[19,47] (Figs. 6 and 7). For the comparisons, the reflectance spectra are converted to emissivity and all spectra are subsequently averaged within the MARA bands, weighted by the respective instrument function (see the sections "MARA Observation Model" and "Comparison of MARA results to spectra of chondrites" under "Methods"). This effective emissivity is weakly temperature dependent, and we evaluate it at 230 K. Considering 300 K and 200 K, the minimum and maximum temperature in the fitted timeframe, results in very minor changes to the band ratios presented in Figs. 4 and 5 and does not affect the conclusion. The spectra in Figs. 5 and 7 are shown as measured, i.e., not normalised or scaled, but shifted by subtracting a constant offset for clarity.

The powdered spectra show systematic differences to the thin sections. The difference of Ryugu's emissivity in the MARA bands is by an order of magnitude larger than the spectral contrast of the spectra of powdered samples and is well matched by the spectra of thin sections. This is evident when comparing Fig. 4, where the ratio of the emissivity within in 9.5–11.5 μm and 13.5–15.5 μm to the one in the 8–9.5 μm are shown, to Fig. 6 where the same is shown for powdered samples and where the estimates of Ryugu differ significantly.

The emissivity in the 5.5–7 μm region drops in most powdered samples and the decrease in emissivity beyond 9 μm is at a longer wavelength in powdered samples compared to thin sections. The latter is due to the different mechanisms behind the emissivity minimum position. In powdered samples, the decrease is due to the combination of fundamental Si–O bond vibrations and transparency features (caused by scattering between the small sample particles), whereas in the thin sections the position of the emissivity minimum is solely representative of the fundamental Si–O bond vibrations, which occur at shorter wavelengths compared to the transparency features. These can be observed comparing Figs. 5a–7.

As shown by the examples of thin-section spectra in Fig. 5a, all thin-section samples show a strong drop in emissivity in the 9.5–11.5 μm band. With an increasing degree of aqueous alteration, the minimum in that band shifts towards shorter

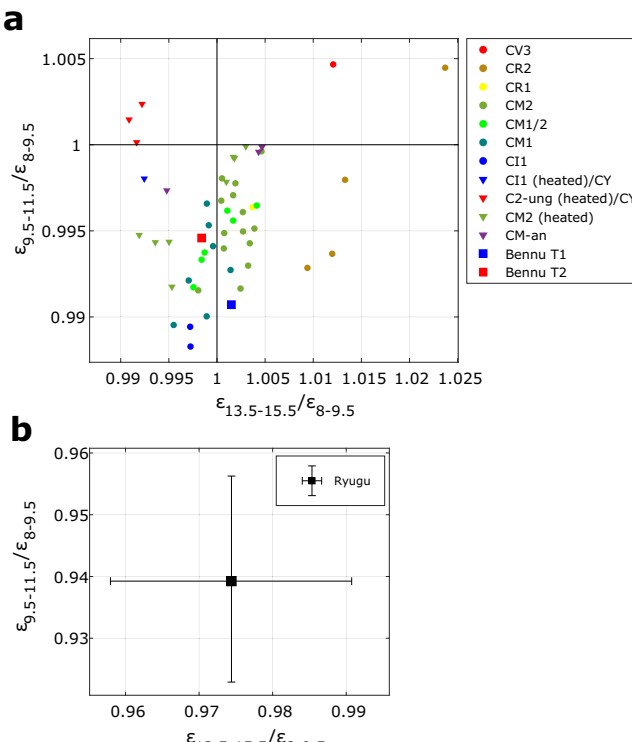

**a**

Legend:
- CV3 (red circle)
- CR2 (orange circle)
- CR1 (yellow circle)
- CM2 (olive circle)
- CM1/2 (green circle)
- CM1 (teal circle)
- CI1 (blue circle)
- CI1 (heated)/CY (blue triangle)
- C2-ung (heated)/CY (red triangle)
- CM2 (heated) (green triangle)
- CM-an (purple triangle)
- Bennu T1 (blue square)
- Bennu T2 (red square)

**b**

Legend:
- Ryugu (black square)

**Fig. 6 Comparison of emissivity ratios derived from MARA data to emissivity ratios derived from powdered chondrite samples and Bennu.** Emissivity spectra are derived from inverted reflectance spectra of powdered samples of carbonaceous chondrites. **a** Ratios of the reflectance and emissivity spectra of various powdered carbonaceous chondrites[19,31,45–47] and OTES Bennu spectra[44], averaged over the instrument function with the 8–9.5 μm, 9.5–11.5-μm and the 13.5–15.5-μm filter channels. The aqueously altered chondrites all show decreasing emissivity from 8–9.5 to 9.5–11.5 μm, whereas the dehydrated CY and the CV3 (Allende) shows an increase in emissivity in that band. **b** The emissivity ratios of Ryugu are significantly lower and far out of bounds compared to panel **a**. Error bars indicate the 25th and 75th percentile of the estimated band ratios. Source data are provided as a Source Data file.

wavelengths, whereas the Christiansen feature shifts towards longer wavelengths[48,49]. The drop increasingly steepens compared to less aqueously altered chondrites where the band is much broader and extends to the neighbouring channels. The emissivity minima of phyllosilicates but also those of olivine lie within the 9.5–11.5 μm band of MARA, such that the mineralogy cannot be distinguished based on the MARA results. However, the MARA bands are sensitive to the broadness and position of the feature. This causes a systematic distinction between the emissivity band ratios of carbonaceous chondrites that were aqueously altered and those that were not. This trend is visible in Fig. 4 where the band emissivity ratios of a variety of carbonaceous chondrites, Ryugu, and the T1 and T2 spectra of Bennu are shown[44]. The band ratios of CM2, CM1/2, CM1, CI1, CR1, Bennu and Ryugu form a cluster of approximate linear shape, from which the other carbonaceous chondrites deviate systematically, i.e., the dry CV3 and CO3 chondrites, thermally metamorphosed chondrites such as the CK4-6, but also the aqueously altered CR2 chondrites. For aqueously altered carbonaceous chondrites, the $\varepsilon_{13.5-15.5}/\varepsilon_{8-9.5}$ ratio is lower compared to the non-aqueously altered ones for a given $\varepsilon_{9.5-11.5}/\varepsilon_{8-9.5}$ ratio. This is caused by a broader emissivity minimum around 11 μm for aqueously altered carbonaceous chondrites (see Fig. 5a). Along this trend, the proportion of the

band ratios is the same, varying only in depth of the bands. This common trend could be caused by the similar shape of mid-IR spectra in that region for chondrites with abundant phyllosilicates, indicating common mineralogy. Among the thin-section spectra, the CI1 meteorites form a cluster closest to the estimated band ratio range for Ryugu. The common trend of aqueously altered chondrites extends to region of powdered samples, which can be seen when comparing the position of the Bennu spectra in Figs. 4–6a. Here, all aqueously altered samples form a cluster in the lower half, pointing towards the band ratio of Ryugu, whereas the CY chondrites and CV3 Allende deviate systematically. The spectra of Bennu shown in Fig. 5b, have significantly lower contrast compared to those of thin-section samples and our estimates for Ryugu, which might be due to fine dust coating the surface of Bennu[44].

## Discussion

The low thermal inertia of the boulders on Ryugu is one of the major surprises of the Hayabusa2 mission, given that based on thermal inertia estimates prior to the arrival of Hayabusa2 a granular surface with millimetre to centimetre-sized regolith was expected to dominate the surface of Ryugu[38]. Bennu's surface, showing a comparable thermal inertia, was expected to be dominated by such regolith as well but, as Ryugu, is dominated by boulders[50]. On Ryugu, the shape of the diurnal temperature variation and in particular the cooling rate at night exclude the presence of a layer of dust thicker than few grains which would result in a drastically different temperature cooling rate[34,39]. Consequently, the low thermal inertia of the boulders can only be explained by high intrinsic porosity. The porosity estimate of $46.7^{+0.3}_{-0.4}\%$ for the observed boulder is considerably higher than those of CM and CI chondrites, i.e., 22% for CM and 34.9% for CI[41]. It is also higher than the porous, pristine fragments of the ungrouped C2 Tagish Lake meteorite, i.e., ~40%[51]. Such high porosity indicates a low strength of the boulder material on both asteroids[34,40,52]. Therefore, fragments of asteroids like Bennu and Ryugu would probably not survive the fall through Earth's atmosphere and representative, macroscopic samples of either asteroid might not be found in meteorite collections.

Another strong hint for the absence of a thick dust layer is the high spectral contrast in the mid-IR emissivity. A dust-dominated surface should show characteristics similar to powdered samples. Yet, the thin-section meteorite spectra are a better match to Ryugu's emissivity. The contrast of the spectra of powdered samples is by an order of magnitude smaller than the MARA estimates, such that powders seem to be a poor analogue to the boulders on Ryugu.

However, the spectra of Bennu and the emissivity estimate of Ryugu both show an emissivity drop in the region around 6 μm[43,44]. Such a drop in emissivity on Bennu is attributed to the presence of a very small amount of fine particles on large rocks, causing volume scattering effects. This drop is less pronounced on Bennu, which may be due to a peak observed near 7 μm that might be missing on Ryugu (see Fig. 5b). It is yet unclear how much dust is required to produce a pronounced minimum around 6 μm on Ryugu without affecting thermal conductivity or obscuring the features in the 8–11 μm region. On Bennu, thermal modelling limits dust to < 50 μm thickness and spectral estimates suggest the presence of only ~5–10 μm of dust on rock surfaces[44,50]. It is furthermore unclear how the scattering effects emerge on such highly porous surface that might be very similar to sintered dust particles[34,40].

Some amount of dust was observed during the sampling operations, where boulders were abraded by the Hayabusa2 spacecraft's thrusters, and around the SCI experiment's impact crater[3,53]. In the

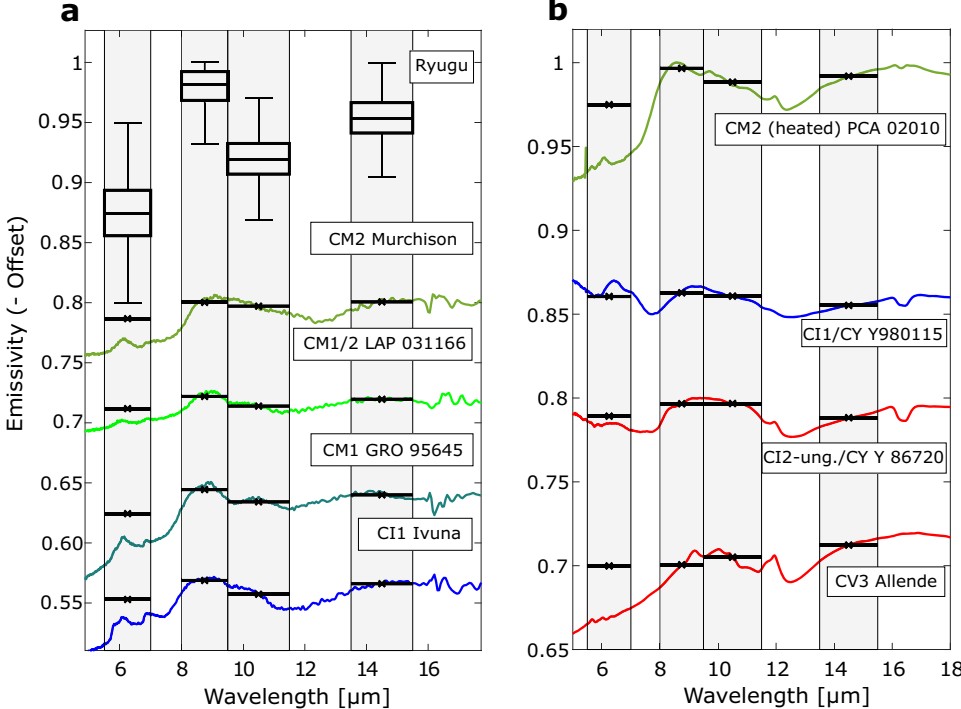

**Fig. 7 Comparison of derived emissivity of Ryugu compared to inverted reflectance spectra of powdered carbonaceous chondrites. a** As in Fig. 5, boxplots of the emissivity estimates for Ryugu in comparison to examples of carbonaceous chondrites, where horizontal lines represent the average emissivity within a MARA band weighted by the instrument function. The four spectra are highly aqueously altered chondrites with increasing degrees of alteration[45]. **b** Emissivity spectra of heated and aqueously altered carbonaceous chondrites[19]. Shown are CM2 PCA 02010 which was exposed to severe heating after being aqueously altered, as well as two CY chondrites, the moderately heated Y 980115 and the severely heated Y 86720. Below is a spectrum of CV3 Allende[47]. The colours of the spectra were chosen to match the corresponding symbols in Fig. 6. Source data are provided as a Source Data file.

latter case, settled fine materials could be identified by a change in albedo around the crater, while showing no change in thermal inertia[40]. Consequently, a very subtle dust layer could also be present on the boulder observed by MARA. Moreover, MARA's 5.5–7 μm channel has a secondary throughput window from 16–20 μm and is sensitive to the spectral features in that region. This can be observed in Figs. 5a and 7a, b where the MARA-equivalent emissivity varies significantly from the measured spectra between 5.5 and 7 μm, which generally leads to lower observed emissivity for solid samples. This ambiguity makes the interpretation of the 5.5–7 μm band emissivity very difficult. Furthermore, the B06 instrument channel, observing the surface in that band, has the highest uncertainty. Since this region is not diagnostic for different groups of carbonaceous chondrites we focus our study on ratios of the other three bands.

The group-level similarity of the mid-IR emissivity to highly aqueously altered CM and CI chondrites strongly supports the presence of phyllosilicates in Ryugu's surface material. This is in line with near-IR observations of the surface gathered with the NIRS3 instrument of the Hayabusa2 main spacecraft which observed a ubiquitous, narrow reflectance minimum at 2.72 μm (3-μm hydration feature) indicating the presence of Mg-rich phyllosilicates, such as antigorite, lizardite, or dehydrated saponite[6,45,54]. The narrowness and weakness of the feature were interpreted as an indication for partial dehydration of Ryugu's surface[6,8] and the absence of other bands in that wavelength range implies that there are no minerals bearing molecular water present at the surface of Ryugu[9]. In contrast, Bennu shows a broader 2.74 μm feature, also indicating the presence of hydrated minerals but perhaps a slightly lesser degree of alteration[43,54]. The estimated mid-IR emissivity of Ryugu shows similarities to the spectra derived on Bennu by the OSIRIS-REx Thermal Emission

Spectrometer (OTES). Although the spectrum of Bennu has less contrast, its general appearance is very similar to that of Ryugu and the emissivity within the MARA bands follows the same trend as the one of CMs, CIs and Ryugu (Fig. 4). The position of emissivity minima and maxima in the mid-IR indicate the presence of phyllosilicates on Bennu[43,44]. This leaves us with the question, why, if the near-IR spectra of Bennu and Ryugu are different due to the partial dehydration postulated for Ryugu, the mid-IR spectra are similar to the same aqueously altered carbonaceous chondrite groups.

Thermogravimetric measurements show that dehydration of phyllosilicates, i.e. the loss of $H_2O$, and dehydroxylation, i.e. the loss of OH, occurs in steps between 200 and 800 °C, depending on the phyllosilicate[55]. The loss of the $H_2O$ and OH groups between the silicate layers leads to the formation of amorphous phyllosilicates. At high temperatures (>700 °C), olivine crystallises changing the structure of the silicates[56]. Some dehydrated phyllosilicate spectra retain a weak 2.72 μm feature[57], as observed in laboratory measurements of moderately dehydrated (above 300 °C but below 700 °C) samples of CI1 chondrite Ivuna[58]. These spectra showed similarity to Ryugu's near-IR spectrum, and indicate that the surface of Ryugu experienced heating within that range[8]. Although the 3-μm absorption in the near-IR is sensitive to the coordination of $H_2O$ and OH, the mid-IR is sensitive to the overall structure of the silicate. Once phyllosilicates are formed, the layered structure of the silicate is to some extent preserved during dehydration as long as the temperature is below the recrystallisation temperature of olivine. This theory is corroborated by mid-IR spectra of moderately heated, aqueously altered, carbonaceous chondrites which retain the phyllosilicate signature in the mid-IR[19,49].

Although it was so far unclear whether the weak 2.72 μm feature was due to partial dehydration of abundant phyllosilicates rather than incomplete aqueous alteration of olivine to phyllosilicates during the formation of Ryugu's parent body, our results show that strong aqueous alteration occurred in Ryugu's parent body prior to any dehydration. The fact the mid-IR emissivity of both Bennu and Ryugu are very similar to the most aqueously altered meteorites in a spectral region that is sensitive to the presence of phyllosilicate, supports the idea that both bodies experienced a high degree of aqueous alteration. Neither Bennu nor Ryugu can be exclusively attributed to either CI or CM on the basis of spacecraft measurements. However, while individual CM chondrites match the emissivity presented here well, the best overall match is obtained by CI chondrites, whereas Bennu's emissivity is slightly better matched by the most aqueously altered CM chondrites[44]. Ryugu could therefore consist of material that was originally more aqueously altered than Bennu.

Recent work found that both asteroids might have originated in the same parent body and formed from sections of the parent body that experienced different extents of heating[59]. However, differences in spectra of bright, presumably exogenic boulders which impacted with the respective parent bodies seem to point towards two distinct parent bodies[60,61]. The temperature limit of <700 °C is consistent with recent studies of potential parent bodies that can explain the observed global distribution of porous rocks on Ryugu indicating a parent body accreting after 2–2.5 Ma after the formation of calcium-aluminium-rich inclusions[40,62,63]. Our results are also consistent with a recent investigation of the 0.7 μm feature in regions of Ryugu that were best shielded against solar heating and space weather[63]. That study found signs of extensive aqueous alteration of the material similar to CM chondrites and subsequent heating below 400 °C.

## Methods

**Thermal model of the MASCOT landing site.** The surface temperature of the MASCOT landing site is calculated by applying a thermophysical model to a digital elevation model (DEM) of the landing site. In this study, a DEM was used which combines the shape model of the boulder in front of the MASCOT lander and a regional DEM[37,64].

The thermophysical model used in this study is based on the previous works[34,35,65]. The surface at each facet of the DEM is considered to be a one-dimensional and homogeneous half-space. The 1D-heat conduction equation is solved:

$$\frac{\partial}{\partial t} T_i(x,t) = \frac{\pi}{\Omega} \frac{\partial^2}{\partial x^2} T_i(x,t) \tag{1}$$

where $\Omega$ is the rotation period, $T_i(x,t)$ is the time- and depth-dependent temperature at facet $i$ with $x$ being depth and with $x=0$ at the surface. The depth is normalised to the diurnal skin depth $d$ which is defined as:

$$d = \sqrt{\frac{k}{c_p \rho} \frac{\Omega}{\pi}} \tag{2}$$

where $\rho$ is the density, $c_p$ specific heat capacity and $k$ thermal conductivity. For this normalisation, $k$, $c_p$, $\rho$ are considered to be constants. For the range of parameters used in this study, the skin depth ranges from 3 to 10 mm, while the finest mesh size for the boulder is 5 mm. The MARA field of view does not resolve the DEM but integrates over a spot of 10–15 cm encompassing more than 3000 facets (Fig. 1b). 3D heat conduction from outside the field of view is negligible on that scale. However, neglecting 3D heat conduction leads to an overestimation of the temperature difference between two adjacent facets and thus an overestimation of the effect of surface roughness on the observed flux. At the same time, the resolution of the DEM does not capture the surface roughness entirely resulting in an underestimation of the surface roughness. Leaving sub-facet roughness as a free parameter as described below, allows for correction of the systematic effect of neglecting 3D heat conduction on the scale of the MARA observation[66].

The temperature profile is calculated for a grid of 100 points equidistantly distributed over total depth of 7.5$d$. The lower boundary defined by setting the flux to zero. The upper boundary condition is defined by the energy balance at the surface:

$$(1-A)I_i(t) = \varepsilon\sigma_B T_i^4(t)\big|_{x=0} + \Gamma\sqrt{\frac{\pi}{\Omega}} \frac{dT_i(t)}{dx}\bigg|_{x=0} - \varepsilon\sigma_B \sum_{j=1}^{N} \nu_{ij} T_j^4(t)\big|_{x=0} \tag{3}$$

where $A$ is the surface bond albedo, $I_i(t)$ is the solar illumination at facet $i$, $\varepsilon$ is the thermal emissivity and $\sigma_B$ is the Stefan–Boltzman constant. Albedo and emissivity are assumed to be identical for all DEM facets. The last term denotes the additional energy received by thermal radiation from the surrounding terrain. It is the sum of the black-body radiation emitted by all other facets $j$ received by facet $i$. The term $\nu_{ij}$ denotes the view factor, i.e., fraction of radiation received by facet $i$ emitted by facet $j$ and the total emitted radiation of facet $j$. If facet $i$ and $j$ are visible to each other, i.e., the line between the facet centres is not obstructed by other facets of the DEM, $\nu_{ij}$ is given by:

$$\nu_{ij} = \frac{a_j \cos\phi_i \cos\phi_j}{\pi\, r_{ij}^2} \tag{4}$$

where $a_j$ is the area of facet $j$, $r_{ij}$ the distance between the facets, $\phi_i$ and $\phi_j$ are the angles between the respective outward normal of facets $i$ and $j$ and the vector connecting their centres. If the line-of-site between the facet centres is obstructed, $\nu_{ij}$ is zero. In this model, the seasonal thermal wave is not considered. Due to the circularity of the orbit, the obliquity of 171.6° (see ref. [2]) and the low thermal inertia this simplification has a negligible influence on the observed temperatures.

The illumination condition was calculated using the NAIF SPICE toolkit. The kernels used are stored in the data repository given below. The calculation was performed starting from October 3, 2018 07:55:00 UTC for one asteroid rotation with $\Omega = 7.632624787382541$ h and divided into 27,477 timesteps, i.e. 1.00002 s per step.

The thermal inertia is defined as $\Gamma = \sqrt{k\rho c_p}$ in units of $J\, m^{-2}\, K^{-1}\, s^{-1/2}$. This parameter is commonly used to describe the amplitude of the diurnal surface temperature variation and its phase shift with respect to maximum insolation. The higher a surface's thermal inertia the later it will reach its maximum temperature in the afternoon, and the smaller is the difference between day and night temperatures.

The surface temperatures of the DEM are pre-calculated varying emissivity and thermal inertia. Here, only the thermal inertia of the facets belonging to the boulder shape model was varied. The thermal inertia of the surrounding terrain facets was interpolated from a global thermal inertia map of Ryugu[42]. Note that this thermal inertia map was obtained assuming an emissivity of 0.9. Higher emissivities result in less than 10% higher thermal inertia. However, increasing the thermal inertia of the landing site by about 10% affects the modelled temperatures by less than the measurement uncertainty such that the uncertainty of the thermal inertia map does not significantly affect the presented results,

**MASCOT sunlight reflection.** The thermal model described above cannot reproduce the steep increase in temperature from sunrise to local noon. MASCOT'S front single-layer insulation (SLI) cover reflects sunlight into the field of view of MARA and increases the energy input there. This effect is accounted for by calculating the incidence angle of the reflected light $\phi_{refl}$ and adding the additional illumination term to the energy balance of the upper boundary condition.

$$I_{refl}(t) = r_{SLI}(1-A)I_0 \cos\phi_{refl}(t) \tag{5}$$

The reflectivity of the SLI layer $r_{SLI}$ is varied between 0 and 100%. The model is an imperfect and exact reproduction of the daytime temperature observation in all six channels is not possible. While a thermal model without the MASCOT reflection can explain afternoon and night-time temperatures without further surface roughness, including the MASCOT reflection increases the modelled temperatures throughout the day and an additional roughness model needs to be included to fit the daytime data. Note that due to the observation geometry, the low emissivity of the MASCOT exterior, and the fact that the SLI cover of the lander reflects the thermal radiation of the surrounding terrain, thermal emission of the lander into the MARA field of view can be neglected in the model.

Sub-facet surface roughness is accounted for by a small-scale crater model[34,67,68]. This model calculates correction factors, dependent on illumination and observation geometry for each of the facets in MARA's field of view. The combined application of reflection modelling and roughness model can produce a near-to-perfect fit in the W10 broadband filter channel but cannot reproduce the observations in the other channels equally well, due to limited accuracy of the shape model and the MASCOT orientation that determine the observation and illumination geometries. Furthermore, the roughness model cannot incorporate two light sources. For this reason, we included only afternoon and night-time observations into our analysis, where the reflection is outside the instrument field of view.

To account for the systematic influence of the model assumptions on the parameter estimation, we consider two models fitting the afternoon and night-time temperature for which very good fits can be obtained. The first model neglects the morning reflection of MASCOT and sub-facet roughness (see Supplement Note 1, Supplementary Figures 1 and 2), the other model includes roughness with the crater density $c$ as a free roughness parameter[34] and assumes 100% reflectivity of the SLI. Despite this range of model assumptions, the results are very similar do not change the conclusion of our study.

**MARA observation model.** The MARA instrument consists of an array of six sensors, observing the surface through different filter windows. Four of the filters are narrow-band filters, two transmit in a broad wavelength range for maximum

signal-to-noise ratio[14,65]. While the thermophysical model calculates the (sub-) surface temperature of the DEM facets, the MARA observation model relates the calculated surface temperatures to the flux received by the individual channels.

The position of the MARA instrument within MASCOT and the orientation and position of MASCOT are provided by NAIF SPICE kernels. The field of view of each MARA channel, was determined during the instrument calibration and found to have an opening angle of 11°.

Using the instrument SPICE kernels and those of the DEM, the facets within each field of view are determined. The view factor between these facets and the respective sensors are calculated using Eq. (4). The observed flux $F_{obs}^i$ in the sensor $i$ is then given by:

$$F_{obs}^i = \sum_{j=1}^{N} \nu_{ij} F_j^i(T_j) \qquad (6)$$

Here $F_j^i$ is the flux received by the sensor $i$ from the facet $j$ with temperature $T_j$ and given by:

$$F_j^i(T_j) = \varepsilon_i \int \tau_i(\lambda) B(T_j, \lambda) \mathrm{d}\lambda \qquad (7)$$

where $B$ is Planck's function, $\tau(\lambda)$ throughput of the filter in front of sensor $i$, $\lambda$ is the wavelength.

The filter throughput is provided in the instrument paper[14] and in the Supplementary Fig. 4. The surface emissivity $\varepsilon_i$ is considered constant within the filter's band. It thus represents an average of the emissivity spectrum with that wavelength, weighted by the Planck's function and the filter throughput[65].

$$\varepsilon_i = \frac{\int \varepsilon(\lambda) \tau_i(\lambda) B(\lambda, T) \mathrm{d}\lambda}{\int \tau_i(\lambda) B(\lambda, T) \mathrm{d}\lambda} \qquad (8)$$

**Ensemble Kalman filter**. The free model parameters are estimated by applying an Ensemble Kalman filter (EnKF)[69] which has been shown to be an effective tool for parameter estimation in thermophysical modelling[35]. The EnKF is a sequential data assimilation method iterating over two steps. At first, a model calculates a forecast of the system's state expressed in vector $\mathbf{z}^f$. Note that lower-case letters in bold font are vectors, upper-case letters in bold font denote matrices. In a second step called analysis, an update of the system's state $\mathbf{z}^a$ is calculated by weighing forecast and observation according to their respective uncertainty. The distributions of analysis and forecast, i.e., posterior and prior distribution, are connected via the likelihood $\ell$ of a partially observed state $\mathbf{y}$ conditioned on the current state estimate $\mathbf{z}^f$. In the case of linear state dynamics and a linear observation model the corresponding distributions are Gaussian distributions $N$ with mean $\mathbf{m}$ and covariance $\mathbf{P}$:

$$N(\mathbf{m}^a, \mathbf{P}^a) \, \ell(\mathbf{y}|\mathbf{m}^f) N(\mathbf{m}^f, \mathbf{P}^f) \qquad (9)$$

Finding the extremum of the posterior $N(\mathbf{m}^a, \mathbf{P}^a)$ leads to the classical Kalman filter for each time instance $\tau_n$:

$$\mathbf{m}^a(\tau_n) = \mathbf{m}^f(\tau_n) - \mathbf{K}(\mathbf{H}\mathbf{m}^f(\tau_n) - \mathbf{y}(\tau_n)) \qquad (10)$$

$$\mathbf{P}^a(\tau_n) = \mathbf{P}^f(\tau_n) - \mathbf{K}\mathbf{H}\mathbf{P}^f(\tau_n) \qquad (11)$$

where $\mathbf{K}$ is the Kalman gain defined as

$$\mathbf{K}(\tau_n) = \frac{\mathbf{P}^f(\tau_n)\mathbf{H}}{\mathbf{R} + \mathbf{H}\mathbf{P}^f(\tau_n)\mathbf{H}} \qquad (12)$$

Here, $\mathbf{H}$ is the observation operator that relates the system's state and observation via $\mathbf{y} = \mathbf{H}\mathbf{z}$ and $\mathbf{R}$ the covariance of the observation.

EnKF is a Monte-Carlo approximation of the classic Kalman filter which has been shown to be accurate even for non-linear systems[70,71]. Here, samples of state variables and parameters are randomly drawn and referred to ensemble members $\mathbf{z}_i$. Mean and covariance are approximated by the empirical mean

$$\bar{\mathbf{m}}^a(\tau_n) = \frac{1}{M} \sum_{i=1}^{M} \mathbf{z}_i^a(\tau_n) \qquad (13)$$

and covariance

$$\bar{\mathbf{P}}^a(\tau_n) = \frac{1}{M-1} \sum_{i=1}^{M} (\mathbf{z}_i^a(\tau_n) - \bar{\mathbf{m}}^a(\tau_n))(\mathbf{z}_i^a(\tau_n) - \bar{\mathbf{m}}^a(\tau_n)) \qquad (14)$$

for each time instance $\tau_n$ and analogously for the prior distribution. The forward model is used to calculate the forecast for each ensemble member based on the analysis from the previous timestep:

$$\mathbf{z}_i^f(\tau_n) = \psi(\mathbf{z}_i^a(\tau_{n-1})) \qquad (15)$$

Where $\psi$ represents the forward model. The analysis step is then performed for each ensemble member via the update matrix $\mathbf{D}$:

$$\mathbf{z}_i^a = \sum_{j=1}^{M} \mathbf{D}_{ij} \mathbf{z}_j^f \qquad (16)$$

The entries of matrix $\mathbf{D}$ depend on the variant of the EnKF applied. In this study an Ensemble Square Root filter is used[72,73]. Here, $\mathbf{D}$ is constructed in a way that for linear systems empirical mean and covariance of the analysis $\bar{\mathbf{m}}^a$ and $\bar{\mathbf{P}}^a$ are identical to the standard Kalman filter analysis.

In this study, an ensemble member is a combination of the state variables and parameters, representing vectors in the augmented state space, i.e., a concatenation of the state space and the parameter space.

$$\mathbf{z}^f = (F_{B06}(t_1), \dots, F_{B06}(t_N), \dots, F_{B13}(t_N), \Gamma, \varepsilon, \varepsilon_{B06}, \varepsilon_{B08}, \varepsilon_{B09}, \varepsilon_{B13}) \qquad (17)$$

Here, the state consists of the six sensor's datasets with $N$ points each equally distribute from early afternoon through the night up to just before sunrise. The free parameters are thermal inertia $\Gamma$, total surface emissivity $\varepsilon$, and the spectral emissivity in the four narrow filter bands of the MARA instrument: $\varepsilon_{B06}, \varepsilon_{B08}, \varepsilon_{B09}, \varepsilon_{B13}$. For the sake of computational efficiency, not all data points are used. Instead, one in every 23 data points corresponding to 11.5 min are used, balancing computational cost and stability of the analysis. The first timestep $t_1$ is set at 13:09 local time (9:42 UTC), and the last timestep is shortly before sunrise.

The forward model $\psi$ for the state, i.e., the modelled MARA observation consists of two parts: the thermal model of the asteroid surface described in section Thermal model of the MASCOT landing site and the MARA instrument model described in section MARA Observation Model. The forward model $\psi$ interpolates the temperatures based on thermal inertia and total emissivity for each ensemble member. With the interpolated temperatures the observation for each MARA channel is calculated using the instrument model. For the four narrow-band filters, the emissivity is the spectral emissivity of the ensemble member. For the two broad filters the emissivity is set to the total emissivity $\varepsilon$.

To be able to traverse the full state space and avoid settling in local minima, a random walk is used.

$$p(\tau_n) = p(\tau_{n-1}) + \zeta \qquad (18)$$

where $p$ represents one of the parameters and $\zeta$ is the inflation parameter. It is chosen such that the standard deviation of the modelled observations based on the distribution of parameter combinations is larger than the mean of the MARA observation uncertainty within the fitted timeframe. The emissivity is forced to be between 0 and 1 and the thermal inertia to be > 0.

**Calibration and uncertainty estimation**. Following the ground calibration of the MARA instrument, it was re-calibrated in flight. For this purpose, a black-body calibration target was situated in front of MARA during the cruise phase when the MASCOT lander was attached to the Hayabusa2 spacecraft. Furthermore, during the descent of MASCOT and during the first part of the surface mission, MARA could observe deep space, which was also used for calibration.

The calibration model describes the voltage in each channel as[14]:

$$U = S(P_t - P_d) + U_{off} + C\frac{\mathrm{d}P_H}{\mathrm{d}T_H} \qquad (19)$$

Three calibration coefficients are used to fit the calibration data: $S$ the sensitivity of the MARA channel's detector in units of [V/W], $U_{off}$ an offset voltage, and $C$ a correction factor weighting the influence of the sensor head heater on the signal voltage $U$. $P_t$ is the radiant power received by the sensor and emitted by the target, $P_d$ the radiant power emitted by the sensor, and $P_H$ is the heating power necessary to stabilise the sensor head temperature $T_H$.

A Monte-Carlo simulation is performed to fit the calibration parameters for each sensor while accounting for uncertainties within the calibration set-up, following the Guide to the Expression of Uncertainty in Measurement (GUM). Four sources of uncertainties were identified, and 10000 random errors were drawn from the corresponding distributions (details provided in Table 1). The calibration parameters are fitted for each realisation. The calibration parameters and their uncertainty are then estimated by taking the mean and standard deviation over the 10,000 realisations.

**MARA data**. The MARA data were collected on October 3, 2018. The brightness temperatures and uncertainties are provided in the source data file. These data are also provided in a Zenodo repository given in "Data availability" section. The repository also contains example datasets for interpretation, verification and extension of the presented work.

The observed flux in each of the MARA instrument filters is calculated from the signal voltage in a two-step process. First the offset voltage $U_{off}$ and the offset generated by the sensor head heat $C\frac{\mathrm{d}P_H}{\mathrm{d}T_H}$ as determined during calibration, are subtracted.

Then the net flux $F_{net}$ on each sensor is calculated using the calibration sensitivity and detector area $A_d$:

$$F_{net} = \frac{U - U_{off} - C\frac{dP_H}{dT_H}}{SA_d} \qquad (20)$$

**Table 1 Sources and distributions of uncertainties considered in the calibration.**

| Uncertainty source | Error estimation |
| --- | --- |
| Thermal and electrical noise of thermopile signal | $N(m_U = 0, \sigma_U = 0.07\mu V)$ |
| Measurement repeatability (depends on the channel) | $N(0, \sigma_{Ur} = 0.16 - 0.5\mu V])$ |
| Calibration uncertainty of calibration target | $\mathscr{U}([-dT_{PT} = -0.13K, dT_{PT} = 0.13K])$ |
| Inhomogeneity of calibration target | $\mathscr{U}([-dT_{cal} = -0.25K, dT_{cal} = 0.25K])$ |

$N$ is Gaussian distribution around mean of thermopile voltage $m_U$ and standard deviation $\sigma_U$. $\mathscr{U}$ is uniform distribution in a given interval. $dU_r$ is difference between the individual in flight calibration measurements under equal conditions, $dT_{PT}$ is the uncertainty of the calibration target's temperature sensor, $dT_{cal}$ is the heterogeneity calibration target's temperature, as observed with an IR camera on the ground.

With the detector temperature known, the flux emitted by the detector $F_{det}$ is calculated. The observed flux is thus:

$$F_{obs} = F_{net} - F_{det} \qquad (21)$$

Uncertainties are estimated from the uncertainty of the signal voltage and the uncertainty of the calibration parameters.

**Porosity estimate based on thermal inertia**. The thermal inertia $\Gamma$ is connected to porosity via

$$\Gamma = \sqrt{k\rho_s(1-\phi)c_p} \qquad (22)$$

$$k = \frac{\Gamma^2}{\rho_s(1-\phi)c_p} \qquad (23)$$

With thermal conductivity $k$, heat capacity $c_p$ and bulk density $\rho_s$. With models for these parameters, the porosity can be calculated from the thermal inertia[34,35,40]. Grain density is set to $\rho_s = 2751$ kg m$^{-3}$[34], heat capacity is given by[38]:

$$c_p = -23.173 + 2.127\,T + 1.5009 \times 10^{-2}T^2 - 7.3699 \times 10^{-5}\,T^3 + 9.6552 \times 10^{-8}\,T^4 \qquad (24)$$

Evaluated at 230 K, which is the average temperature in the considered timeframe. For the thermal conductivity, we use a model based on thermal conductivity measurements of highly porous aggregates of micrometre-sized silicate particles with porosities ranging from 40 to 90%[74]:

$$k_m = 3.3\left[\exp(-50\phi) + \exp(-4.4 - 23.5\phi)\right]^{1/4} \qquad (26)$$

This model has been applied to estimate the porosity of anomalous, highly porous boulders on Ryugu[40]. The porosity is then estimated from thermal inertia by solving $k = k_m$ for $\phi$. Based on thermal inertia of $256^{+4}_{-3}$ J m$^{-2}$ K$^{-1}$ s$^{-1/2}$, we obtain a porosity of $46.7^{+0.3}_{-0.4}$%.

A second model which has been applied Ryugu in previous works[34,36,40] can explain the thermal conductivity of meteorites well up to porosities of 25% as the porosity of meteorites for which thermal conductivity has been measured is typically lower[41]. The model needs to be extrapolated far beyond this range to be applied to Ryugu and we consider it therefore to be less suitable. Applying this model would result in higher porosity of $56.2^{+0.3}_{-0.5}$%.

**Comparison of MARA results to spectra of chondrites**. The presented emissivity estimates are compared to mid-IR spectra of various chondrites. These are reflectance spectra $R(\lambda)$ that are converted to emissivity via Kirchhoff's law $\varepsilon(\lambda) = 1 - R(\lambda)$. We collected data from multiple studies, including spectral data from literature of powdered meteorite samples[19,31,45–47], meteorite thin sections[44] and OTES observations of Bennu[44], as well as previously unpublished data by V.E. Hamilton, and A. Maturilli. All data presented in this study are included in the Supplementary Source Data file.

In Bates et al.'s study[45], mid-IR (5–30 µm) reflectance spectra were measured using a Bruker VERTEX 70 V FTIR in a diffuse reflectance geometry. Samples were heated at 150 °C for 2 h to remove adsorbed water. Measurements were then conducted under vacuum (2 hPa) with spectral resolution of 4 cm$^{-1}$. A wide range mid to far-IR beamsplitter was used in combination with deuterated L-alanine doped triglycine sulfate (RT-DLaTGS) detector at room temperature. Each of the meteorite spectra were calibrated using a diffuse gold calibration target measured under the same conditions.

In Bates et al.'s study[19], mid-IR (5.5–50 µm) emissivity spectra were measured using the Planetary Analogue Surface Chamber for Asteroid and Lunar Environments (PASCALE)[31]. Under SAE conditions, the near-surface environment of an airless body is simulated. Measurement are conducted under vacuum (<10$^{-4}$ mbar), while cooling the interior of the chamber to <150 °C using liquid N$_2$. A thermal gradient in the upper hundreds of microns of the sample was induced by heating the samples from above by a quartz-halogen lamp attached to the outside of the chamber, and below by a sample cup heater until a maximum brightness temperature of the sample of ~75 °C[31] is achieved. To obtain the instrument response function and to calibrate the data to effective emissivity, daily

measurements of a black-body target were made at two temperatures that bracket the sample temperature (~67 °C and ~87 °C).

Reflectance spectra in Beck et al.[46] and further spectra collected by A. Maturilli[47], were acquired under vacuum at the Planetary Spectroscopy Laboratory (PSL) at DLR Berlin. For measurements, a Bruker Vertex 80 V was used under vacuum at <1 mbar. The measurements were conducted in an air-conditioned room. The bi-directional reflectance of the samples was measured varying incidence and emission angles between 13 and 85°. The viewing cone had an aperture of 17°, small enough to define those measurements as bi-directional[75]. As a light source, a high-power Globar lamp (24 V, water-cooled) is used covering the MIR spectral range. For each session, reflectance spectra are calibrated against an Infragold certified reference.

Reflectance measurements of carbonaceous chondrite samples prepared as thin sections[44] were collected under ambient conditions using a Thermo Scientific iN10 Fourier transform infrared microscope (µ-FTIR) equipped with an MCT-B detector and KBr beamsplitter with a spectral range from 4000–400 cm$^{-1}$ (2.5–25 µm). The instrument's ×15, 0.7 N.A. (half angle range 20° to 43.5°) visible/IR objective and condenser are permanently aligned. This set-up results in a sufficiently small solid angle preventing band broadening effects near the Christiansen feature (CF) observed in biconical systems if solid angles are large, including (near-) grazing angles of incidence and collection. Radiance spectra are rated to a polished gold plate to obtain reflectance[76]. Spectral maps are collected on polished thin sections at a spectral resolution of 4 cm$^{-1}$, spot size of 300 µm and step size of 300 µm. To obtain the bulk spectrum of a samples, map spectra are averaged.

To compare these spectra to the observations of MARA, the spectra $\varepsilon(\lambda)$ are averaged within the spectral bands of MARA, weighted by the instrument function according to Eq. (7), resulting in an effective emissivity that MARA would obtain in each filter.

## Data availability
The reported MARA data, i.e. brightness temperatures and uncertainties, are provided included in the supplementary source data file and are also available in Zenodo repository 5796141 with the https://doi.org/10.5281/zenodo.5796141. The repository also contains an example dataset to reproduce and extend the work presented here, as well as the MASCAM image shown in Fig. 1a. The SPICE Kernels used in this work, including the DEM of boulder and landing site, are available in the same repository. Furthermore, the presented analysis includes spectral data from literature of powdered meteorite samples[19,31,45–47], meteorite thin sections[44] and OTES observations of Bennu[44]. All data presented in this work, including previously unpublished spectral data, are included in the supplementary source data file. Source data are provided with this paper.

## Code availability
The codes for the Data Assimilation of the MARA data and the comparison to meteorite spectra are available in the Zenodo repository 5796141, with the https://doi.org/10.5281/zenodo.5796141. The code for the thermal model is available from the corresponding author upon reasonable request.

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

## Acknowledgements

M.H. was financially supported by Geo.X, grant number: SO_087_GeoX. M.H. and J.d.W. were financially supported by Deutsche Forschungsgemeinschaft (DFG)—SFB1294/1–318763901. W.N. acknowledges support by Klaus Tschira Foundation. This study was partially supported by JSPS Core-to-Core programme "International Network of Planetary Sciences".

## Author contributions

M.H., M.G., J.K. and T.-M.H. collected, calibrated and interpreted the MARA dataset. H.S., F.S., K.D.M., Y.S., N.S., W.N., S.T., T.O., F.P., S.E., E.K. and R.J. performed and contributed to the thermal modelling of Ryugu's surface using the combined digital elevation model of boulder and landing site and interpreted retrieved thermophysical properties J.d.W. contributed to the parameter estimation and the implementation of the data assimilation method. V.E.H., H.B., A.M., J.H. and S.S. interpreted Ryugu's mid-IR emissivity and compared the results to spectra of various meteorites and Bennu. All co-authors discussed the results of this study and contributed to the preparation of this manuscript.

## Competing interests

The authors declare no competing interests.
