## [Peer Review File · Nature Communications]

REVIEWER COMMENTS

Reviewer #1 (Remarks to the Author):

This paper presents a detailed thermophysical analysis of the boulder observed by the MASCOT lander on asteroid Ryugu. It is similar to the authors' previously published work (Grott et al. 2019; Hamm et al. 2020), but the main improvements are analysis of multiple infrared (IR) data channels, use of a digital elevation model (DEM) in the temperature calculations, and the first in-situ derivation of Ryugu's mid-IR emissivity spectrum. The latter is the most notable new result from this work, as it demonstrates that Ryugu is compositionally similar to asteroid Bennu and CM/CI meteorites with evidence of aqueous alteration. This result is important as it provides context for the laboratory analysis of the samples recently returned from Ryugu by Hayabusa2. Overall, the paper is well written, figures are nice and appropriate, and the quoted results have suitable and reasonable uncertainties associated with them. However, I have several minor comments that need addressing before the paper is suitable for publication (detailed below).

Minor comments:

- Lines 55-56: The statement that MARA provides the only mid-IR spectral data of Ryugu is not quite true. The Spitzer space telescope also acquired a mid-IR (5 to 38 μm) spectrum of Ryugu, although it was disk-integrated (Campins et al. 2009). This statement should therefore be clarified, and the authors should consider comparing their results to the emissivity spectrum derived from Spitzer. However, I suspect that the Spitzer data would be too noisy for a detailed comparison, but there does seem to be a feature at $\sim 15 \mu\text{m}$ in the Spitzer spectrum that appears inconsistent with the MARA results in its B13 channel.

- Lines 76-77: The authors should justify this statement as to why it was not feasible to do the multiple IR channel analysis using their previous models. Ultimately, it is the same dataset being analysed here, and the new DEM model gives the same thermal inertia result as their previous work (Grott et al. 2019; Hamm et al. 2020). It would be useful to compare the temperature predictions from the DEM model to the predictions of their previous models, as the improvement in the W10 channel is not immediately obvious and justified to the reader.

- Lines 347-351: The mesh-size of the DEM used should be stated here and compared against Ryugu's diurnal thermal skin depth to justify the use of the 1-D heat conduction simplification. Can you really neglect 3-D heat conduction in this modelling scenario?

- Lines 358-359: This seems to imply that Ryugu's seasonal thermal wave is also neglected in the temperature calculations. If so, then this should be specified here and justified. I suspect that Ryugu's moderately circular orbit results in a minimal seasonal thermal wave.

- Line 375: A supplementary table summarising the input dates/times and Ryugu geometries would be a useful addition to describe what conditions the DEM surface temperatures were pre-calculated for.

- Lines 376-377: The global thermal inertia map of Ryugu derived by Shimaki et al. (2020) was produced assuming an emissivity of 0.9, but the emissivity derived in this study was ~ 0.98 . How does the difference in emissivity impact the results of Shimaki et al. (2020), and the resulting application of its thermal inertia for modelling the terrain surrounding the MASCOT boulder?

- Lines 378-403: This section nicely considers the potential impact of reflected sunlight from the MASCOT lander on the model results, but what about thermal radiation from the lander too? What is the lander's temperature, and is it a significant source of input energy for the model?

References:

Campins et al. 2009, A&A, 503, L17

Grott et al. 2019, Nature Astronomy, 3, 971

Hamm et al. 2020, MNRAS, 496, 2776

Shimaki et al. 2020, Icarus, 348, 113835

Reviewer #2 (Remarks to the Author):

In the manuscript, new in situ observations of Ryugu's surface from the MARA thermal infrared instrument onboard MASCOT are discussed and compared against laboratory measurements of analog meteorites. Results from these comparisons show that the observed boulder, which is common to the other boulders on Ryugu, has a thermal inertia consistent with high porosities and spectra that are consistent with whole rock reflectance spectra suggesting that fine particulate dust

is not contributing to the low thermal inertia signal. Spectral results also show that Ryugu's boulders have been aqueously altered. MARA's in situ observations of Ryugu's surface are a first of their kind and are of great interest to the community.

While the comparisons seem standard for the community and sound, the MARA compositional filter transmission curves are not provided. Thus, the re-sampling of laboratory meteorite spectra cannot be re-produced by the community.

Additional details below should be addressed:

Lines 62-64: It might be important to note that the CF is also indicative of the bulk composition of whatever mineral/rock is being investigated.

Lines 64-65: Need to include a few references to work that has been done to show the use of these diagnostic vibration bands.

Lines 65-67: Water also has a diagnostic absorption band in the 4-7 micron region and should be mentioned here with the appropriate reference. A lack of feature in this region would further support the idea that the asteroid has become dehydrated. Also, you should add the word 'as' between 'such' and 'pyroxene'.

Lines 68-69: It is not clear what is meant by 'The minima and maxima are stronger under vacuum'. Are you meaning that the spectral contrast of the features increases under vacuum? Or are you trying to say something else? If you look at lab studies, it is only the spectral contrast between the CF and the first vibration that you see increase in vacuum conditions. Also, it is confusing to mix particle size effects with vacuum effects here. I would suggest separating the two.

Lines 69-72: It is not clear what is meant by 'contrast is reduced'. Only the vibration bands decrease in spectral contrast as particle size decreases. Need to be specific here.

Lines 145 – 147: I think you mean that all the spectra, once they are in emissivity, are re-sampled using the MARA band filters, but as written it reads like only the reflectance spectra are being re-sampled.

Lines 167 – 169: Here in the text and in the caption for figure 4 it is stated that there are systematic differences between aqueously altered materials and anhydrous materials. However, the systematics and what is driving those systematics are not discussed.

Figure 4: The emissivity plots b and c are offset for clarity, but the y-axis label does not indicate that rather this information is buried within the long caption.

Figure 5: It is not clear why Allende is plotted in this figure especially if spectra of the other anhydrous spectra are not being included.

REVIEWER COMMENTS

Reviewer #1 (Remarks to the Author):

This paper presents a detailed thermophysical analysis of the boulder observed by the MASCOT lander on asteroid Ryugu. It is similar to the authors' previously published work (Grott et al. 2019; Hamm et al. 2020), but the main improvements are analysis of multiple infrared (IR) data channels, use of a digital elevation model (DEM) in the temperature calculations, and the first in-situ derivation of Ryugu's mid-IR emissivity spectrum. The latter is the most notable new result from this work, as it demonstrates that Ryugu is compositionally similar to asteroid Bennu and CM/CI meteorites with evidence of aqueous alteration. This result is important as it provides context for the laboratory analysis of the samples recently returned from Ryugu by Hayabusa2. Overall, the paper is well written, figures are nice and appropriate, and the quoted results have suitable and reasonable uncertainties associated with them. However, I have several minor comments that need addressing before the paper is suitable for publication (detailed below).

Thank you very much for your positive feedback and the very helpful comments that we address below.

Minor comments:

- Lines 55-56: The statement that MARA provides the only mid-IR spectral data of Ryugu is not quite true. The Spitzer space telescope also acquired a mid-IR (5 to 38 μm) spectrum of Ryugu, although it was disk-integrated (Campins et al. 2009). This statement should therefore be clarified, and the authors should consider comparing their results to the emissivity spectrum derived from Spitzer. However, I suspect that the Spitzer data would be too noisy for a detailed comparison, but there does seem to be a feature at $\sim 15 \mu\text{m}$ in the Spitzer spectrum that appears inconsistent with the MARA results in its B13 channel.

Yes, this is true, we meant to say that MARA is the only spectrally resolved mid-IR data of the mission and clarify this in the revised manuscript.

Comparison of our results with the Spitzer data is on the one hand hindered by the noise of the Spitzer observation, and on the other hand that comparison of a disk-integrated observation to that of a single boulder might be misleading. At that scale the spectrally distinct parts of Ryugu are mixed. For asteroid (101955) Bennu, Hamilton et al., 2019 report disk-integrated spectra with similar signal-to-noise ratio as the Spitzer observation of Ryugu. Later, spatially resolved data reported in Hamilton et al. 2021, showed two spectrally distinct types of material that were not observed in the disk-integrated observation. Furthermore, the shape and rotation axis of Ryugu turned out to be different from the one assumed in Campins et al., 2009. Finally, comparison to the B13 results is hindered by a gap in the Spitzer data at 13 μm .

- Lines 76-77: The authors should justify this statement as to why it was not feasible to do the multiple IR channel analysis using their previous models. Ultimately, it is the same dataset being analysed here, and the new DEM model gives the same thermal inertia result as their previous work (Grott et al. 2019; Hamm et al. 2020). It would be useful to compare the temperature predictions from the DEM model to the predictions of their previous models, as the improvement in the W10 channel is not immediately obvious and justified to the reader.

The analysis in Grott et al., 2019 and Hamm et al., 2020, only included the W10 channel, while the presented work discusses the full data set for the first time. It is true that the thermal inertia result is effectively the same as before. However, the thermal inertia is not the main outcome in this study. The main results are the emissivity estimates in the narrow band filters. Without the DEM, the orientation of the surface and the view factor to the surrounding terrain had to be treated as a free parameter which drastically increased the uncertainty of the other parameters, hindering a meaningful estimation of the narrow band emissivity. We agree that we need to clarify this in the manuscript and add the following:

“In previous works, the unknown orientation of the observed surface and the thermal radiation received from the surrounding terrain had to be included as free parameters in the analysis which increased the uncertainty of the emissivity estimate within the narrow-band filters to a degree that hindered a meaningful interpretation. With the derivation of a detailed 3D shape model of the boulder based on MASCOT camera images (Scholten et al. 2019) (Fig. 1), the orientation of the observed surface as well as its radiative heat exchange with the local terrain can be considered fixed and the analysis of the full MARA dataset is feasible.”

Concerning the comparison of the temperature predictions we add the figure below to the supplement. The previous results are similar to the reduced model. The nighttime temperatures are equally well predicted. However, a major improvement is obtained in the early morning temperatures, highlighting the necessity to include the reflected sunlight into the analysis. We stress this aspect in the manuscript by adding the following to:

l. 130-132 “The results for the simplified model are summarized in the supplementary information along with a comparison to earlier studies (Grott et al., 2019) which are very similar to the reduced model.”

Supplement:

“Fig. S3 shows the comparison between the full model shown in the main article and previous works (Grott et al., 2019) for the W10 channel. The figure highlights the improvement around sunrise and also shows the similarity between the reduced model and previous works. The reduced model (Fig. S1) is almost identical to the previous studies but with surface orientation and radiative heat exchange with the surrounding being fixed. Further, surface roughness is also included to some degree via the boulder DEM”

Figure S3: Temperature prediction of the full model compared to earlier results. Observed brightness temperature (black) in the W10 band is shown as a function of local time with grey indicating the 2σ uncertainty. The Solid red line represents the temperature prediction of the full thermal model based on the first, second, and third quartile of estimated model parameters, blue solid lines represent the temperature prediction based on the outliers of the parameter estimation. The results from Grott et al. (2019) are shown for comparison, with the red dashed line representing the best-fitting model prediction without roughness and the green solid line representing the model prediction including roughness correction for the daytime temperatures. The presented full model explains the daytime temperatures compared to earlier simplified models better.

- Lines 347-351: The mesh-size of the DEM used should be stated here and compared against Ryugu's diurnal thermal skin depth to justify the use of the 1-D heat conduction simplification. Can you really neglect 3-D heat conduction in this modelling scenario?

We agree and add the mesh size to the method section and compare it to the diurnal skin depth. The mesh size of the boulder model is 5mm while the skin depth is 5 – 10 mm depending on the assumed thermal inertia. This shows that 1D-heat conduction is indeed a simplification resulting in an overestimation of the temperature differences between two adjacent facets of the DEM (Davidsson and Rickman, 2014).

However, the MARA observation integrates over 3000 facets in its field of view such that the observation corresponds to a spot of roughly 10-15 cm in size. At that scale the 3D heat-conduction is neglectable, and areas adjacent to the MARA field of view will not influence the measurement significantly via heat

conduction. As reported by Davidsson and Rickman, 2014, 3D-heat conduction below the resolution of the observation mainly influences the roughness correction factor and can be captured by 1D-heat conduction models nevertheless.

Using a DEM, rather than a homogenous half-space as in previous works, has three effects: It fixes the average surface orientation within the MARA field of view, it fixes the infrared radiation from its surrounding terrain (the facet size of the terrain model ranges from 50 cm to 10 m), and it accounts for roughness as the DEM resolution is smaller than the MARA field of view. Neglecting 3D heat conduction mainly affects the last aspect and for a perfect DEM it would result in an overestimation of the roughness effect. However, in contrast to the neglect of 3D-heat conduction, the limited resolution of the DEM causes an opposing systematic effect. Compared to the real surface the DEM is much smoother, and cannot capture the rugged, cauliflower appearance of the boulder surface and therefore underestimates the roughness of the surface. The fact that we need to introduce additional sub-facet roughness correction to best explain the observation indicates that the 1D thermal model of the boulder DEM still underestimates thermal contrasts within the field of view rather than overestimating them. We add this justification to the method section.

- Lines 358-359: This seems to imply that Ryugu's seasonal thermal wave is also neglected in the temperature calculations. If so, then this should be specified here and justified. I suspect that Ryugu's moderately circular orbit results in a minimal seasonal thermal wave.

Yes, due to the moderately circular orbit, the obliquity of 171.6° (Watanabe et al., 2019), and also due to the low thermal inertia, the effect of the seasonal thermal wave can be neglected. We specify and justify it in the methods.

We ran a simplified thermal model assuming a spherical shape of Ryugu, with the observed spin axis and orbit properties, once with a fixed position on an orbit corresponding to the MARA observation epoch and once integrating over orbits. The difference depends slightly on the thermal inertia and up to $400 \text{ J m}^{-2} \text{ K}^{-1} \text{ s}^{-1/2}$ it is below 1 K and thus below the uncertainty of the MARA measurements.

- Line 375: A supplementary table summarizing the input dates/times and Ryugu geometries would be a useful addition to describe what conditions the DEM surface temperatures were pre-calculated for.

The geometry and illumination conditions were calculated using the NAIF SPICE toolbox. The corresponding SPICE Kernels will be provided in a public repository (<https://doi.org/10.5281/zenodo.5679590>) and referred to in the Data availability statement. We add a paragraph providing the starting date and timestep of the simulation.

- Lines 376-377: The global thermal inertia map of Ryugu derived by Shimaki et al. (2020) was produced assuming an emissivity of 0.9, but the emissivity derived in this study was ~0.98. How does the difference in emissivity impact the results of Shimaki et al. (2020), and the resulting application of its thermal inertia

for modelling the terrain surrounding the MASCOT boulder?

The thermal inertia estimate of the map derived in Shimaki et al. (2020) is only weakly depending on the assumed emissivity. Shimaki et al. (private communication) found that, e.g., an emissivity of 0.95 results in a less than 7% higher average thermal inertia estimate. At the same time, the field of view of MARA is most influenced by the boulder itself with variable thermal inertia. The view factor to the rest of the landing site is relatively small, < 5% such that changes in the thermal inertia there have only a negligible influence on the observed infrared flux. We found that systematically increasing thermal inertia of the landing site by $20 \text{ Jm}^{-2}\text{K}^{-1}\text{s}^{-1/2}$ results in less than 0.15 K higher nighttime temperatures, and less than 0.1 K lower daytime temperatures. Since this is smaller than the observation uncertainty we consider this systematic effect negligible for the parameter estimation and interpretation thereof.

We add a clarification to the end of that method subsection.

- Lines 378-403: This section nicely considers the potential impact of reflected sunlight from the MASCOT lander on the model results, but what about thermal radiation from the lander too? What is the lander's temperature, and is it a significant source of input energy for the model?

Due to the low emissivity of the MASCOT exterior and the small view factor of the MARA field of view to MASCOT which sits partially below the front edge of the boulder, the thermal influence is small. MASCOT radiates its heat via radiators that are mounted on the top of the lander, facing open space. The SLI cover facing the boulder mainly reflects the IR radiation of the surrounding. Based on this, the observation geometry, and temperatures of the MASCOT sub-systems between 260 and 320 K (Cozzoni et al., 2021), we estimate that the thermal disturbance by MASCOT is below the uncertainty of the temperature measurement. This is confirmed by the fact that the sudden increase in the subsystem temperatures reported in Cozzoni et al., 2021, e.g., the camera electronics when illuminating the surface with LEDs and taking images, do not show up in the MARA data.

We add a short statement to this method subsection

Reviewer #2 (Remarks to the Author):

In the manuscript, new in situ observations of Ryugu's surface from the MARA thermal infrared instrument onboard MASCOT are discussed and compared against laboratory measurements of analog meteorites. Results from these comparisons show that the observed boulder, which is common to the other boulders on Ryugu, has a thermal inertia consistent with high porosities and spectra that are consistent with whole rock reflectance spectra suggesting that fine particulate dust is not contributing to the low thermal inertia signal. Spectral results also show that Ryugu's boulders have been aqueously altered. MARA's in situ observations of Ryugu's surface are a first of their kind and are of great interest to the community.

While the comparisons seem standard for the community and sound, the MARA compositional filter transmission curves are not provided. Thus, the re-sampling of laboratory meteorite spectra cannot be re-produced by the community.

We thank you for your encouraging feedback and your constructive comments. We addressed the issues below. Concerning the reproducibility, we agree and provide the throughputs along with the public code repository (<https://doi.org/10.5281/zenodo.5679590>) to facilitate reproducibility (Filter_throughput.mat in the SpectralPlots.zip downloadable from that repository). That repository contains the MATLAB functions necessary to reproduce our work. We also add a figure with the narrow-band filter throughputs to the supplement.

Figure S4: Throughput of the four narrowband filter. Throughput, i.e. combination of transmissivity of the filter and absorption of the sensor absorber, is plotted against wavelength. Secondary throughput windows appear at wavelength larger 15 μm and are most pronounced in the B06 filter.

Additional details below should be addressed:

Lines 62-64: It might be important to note that the CF is also indicative of the bulk composition of whatever mineral/rock is being investigated.

Included as proposed

Lines 64-65: Need to include a few references to work that has been done to show the use of these diagnostic vibration bands.

We include references to Lyon, 1965, Conel, 1969, Sandford, 1984, Salisbury and Walter, 1989, Salisbury et al., 1991b, and Bates et al., 2021.

Lines 65-67: Water also has a diagnostic absorption band in the 4-7 micron region and should be mentioned here with the appropriate reference. A lack of feature in this region would further support the idea that the asteroid has become dehydrated. Also, you should add the word 'as' between 'such' and 'pyroxene'.

We add this aspect to the paragraph and refer to Aines and Rossman (1984), Salisbury et al. (1991a). We add the word 'as' as proposed.

Lines 68-69: It is not clear what is meant by 'The minima and maxima are stronger under vacuum'. Are you meaning that the spectral contrast of the features increases under vacuum? Or are you trying to say something else? If you look at lab studies, it is only the spectral contrast between the CF and the first vibration that you see increase in vacuum conditions. Also, it is confusing to mix particle size effects with vacuum effects here. I would suggest separating the two.

We agree and rewrite that part to:

"The mid-IR spectrum also depends on physical conditions. As the particle size becomes comparable to the wavelength of the measurement, the contrast of the vibrational modes is reduced and new features, referred to as transparency features, appear due to volume scattering. In silicates, transparency features occur at wavelengths shorter than the CF and in the interband region between ~11 – 13 μm as well as at longer wavelength (Aronson and Emslie, 1973, Moersch and Christensen, 1995, Le Bras and Erard, 2003). Conversely, in fine, transparent particulate materials, steep thermal gradients can develop in the upper few 10s to 100s of μm and increase spectral contrast due to the measurement of multiple temperatures simultaneously. This effect is enhanced under very low pressure and vacuum conditions (Logan et al. 1973, Salisbury and Walter 1989). However, for low albedo materials, including carbonaceous chondrites containing opaques and insoluble organic material, this effect is substantially less pronounced (Henderson and Jakosky, 1994; 1997; Salisbury et al., 1991b, Donaldson Hanna et al., 2021). On Ryugu, with its dark and dust-deficient surface volume scattering and thermal gradients should contribute only little to the observed emissivity."

Lines 69-72: It is not clear what is meant by 'contrast is reduced'. Only the vibration bands decrease in spectral contrast as particle size decreases. Need to be specific here.

We agree and change that lines along with the comment above to be more specific.

Lines 145 – 147: I think you mean that all the spectra, once they are in emissivity, are re-sampled using the MARA band filters, but as written it reads like only the reflectance spectra are being re-sampled.

Changed as proposed.

Lines 167 – 169: Here in the text and in the caption for figure 4 it is stated that there are systematic differences between aqueously altered materials and anhydrous materials. However, the systematics and what is driving those systematics are not discussed.

The systematic differences in the MARA band ratios between the aqueously altered carbonaceous chondrites and the others is mostly caused by the broadness of the emissivity minimum between 10 and 13 μm . To explain this better we change l. 167-169 (in revised version l 180-182) to:

“However, the MARA bands are sensitive to the broadness and position of the feature. This causes systematic distinction between the emissivity band ratios of carbonaceous chondrites that were aqueously altered and those that were not. This trend is visible in Fig.4a ...”

We furthermore add a brief description of the main systematic distinction to the paragraph:

“For aqueously altered carbonaceous chondrites, the $\epsilon_{13.5-15.5}/\epsilon_{8-9.5}$ ratio is lower compared to the non-aqueously altered ones for a given $\epsilon_{9.5-11.5}/\epsilon_{8-9.5}$ ratio. This is caused by a broader emissivity minimum around 11 μm for aqueously altered carbonaceous chondrites (s. Fig. 4b).”

A discussion of the more fundamental reasons for the difference in the silicate bands is given in the paragraph l. 248 – 253 (265-277 in revised manuscript).

Figure 4: The emissivity plots b and c are offset for clarity, but the y-axis label does not indicate that rather this information is buried within the long caption.

Changed as proposed, here and in figure 5.

Figure 5: It is not clear why Allende is plotted in this figure especially if spectra of the other anhydrous spectra are not being included.

Allende is a widely known meteorite and we have spectra of solid and powdered samples. We therefore assumed that it would be a good example to show the strong differences of CV to CM and CI chondrites as well as systematic differences between powdered and solid samples.

REFERENCES:

Hamilton, V.E., et al. (2019) Nature Astronomy 3, 332 - 340.

Hamilton, V.E., et al. (2021). Astron. Astrophys., 650, A120.

Grott et al. 2019, Nature Astronomy, 3, 971

Hamm et al. 2020, MNRAS, 496, 2776

Scholten et al., 2019, Astron. Astrophys., 632, L5

Davidsson and Rickman, (2014), Icarus, 243, 58-77

Watanabe et al., (2019), Science, 10.1126/science.aav8032

Shimaki et al. 2020, Icarus, 348, 113835

Cozzoni et al., (2021) PSS, 205, 105286

Grott et al., (2017), SSR, 208, 413-431

Lyon, R.J.P. (1965), Econ. Geol. 60, 715-736.

Conel, J.E. (1969), J. Geophys. Res. 74, 1614-1634

Sandford, S.A. (1984), Icarus 60, 115-126.

Salisbury and Walter(1989) J. Geophys. Res. 94, 9192-9202

Salisbury, et al. (1991b), Icarus 92, 280-297

Bates et al., (2021), JGR:Planets, e2021JE006827

Aines and Rossman (1984) J. Geophys. Res. 89, 4059-4071

Salisbury, et al. (1991a) Infrared (2.1-25 μm) Spectra of Minerals. The Johns Hopkins University Press, Baltimore and London.

Aronson and Emslie (1973) Appl. Opt. 12, 2573-2584.

Moersch and Christensen (1995) J. Geophys. Res. 100, 7,465-467,477

Le Bras and Erard (2003), Plan. Space Sci. 51, 281-294

Logan, et al. (1973) J. Geophys. Res. 78, 4983-5003

Henderson, B.G. and Jakosky, B.M. (1994) J. Geophys. Res. 99, 19063-19073.

Henderson, B.G. and Jakosky, B.M. (1997). J. Geophys. Res. 102, 6567-6580

Donaldson et al. (2021). JGR: Planets, 126, e2020JE006624

REVIEWERS' COMMENTS

Reviewer #1 (Remarks to the Author):

I am happy with the changes and responses provided by the authors, and I have no further comments.

Reviewer #2 (Remarks to the Author):

I believe the revised manuscript has adequately addressed all of the concerns that were raised in the initial draft and is ready for publication.

A few minor suggested edits in the marked up manuscript (track changes file):

Line 76 - I would suggest replacing 'Conversely' with 'Additionally' or something to that affect as I don't think you actually mean conversely.

Line 78 - It is a bit more nuanced than measuring multiple temperatures simultaneously, it is that these temperatures come from different depths within the sample as the optical properties vary across the MIR spectrum.

Line 81 - I would add a comma between surface and volume.

Line 81 - I would remove the word 'only' between contribute and little.

Response to Review:

Reviewer #1 (Remarks to the Author):

I am happy with the changes and responses provided by the authors, and I have no further comments.

Thank you very much for your constructive review which helped us to improve our manuscript.

Reviewer #2 (Remarks to the Author):

I believe the revised manuscript has adequately addressed all of the concerns that were raised in the initial draft and is ready for publication.

Thank you very much for you positive feedback.

A few minor suggested edits in the marked up manuscript (track changes file):

Line 76 - I would suggest replacing 'Conversely' with 'Additionally' or something to that affect as I don't think you actually mean conversely.

Changed as suggested.

Line 78 - It is a bit more nuanced than measuring multiple temperatures simultaneously, it is that these temperatures come from different depths within the sample as the optical properties vary across the MIR spectrum.

We add this clarification to the line.

Line 81 - I would add a comma between surface and volume.

Added as proposed

Line 81 - I would remove the word 'only' between contribute and little.

Removed as proposed.